# DNAct: Diffusion Guided Multi-Task 3D Policy Learning

## Abstract

This paper presents DNAct, a language-conditioned multi-task policy framework that integrates neural rendering pre-training and diffusion training to enforce multi-modality learning in action sequence spaces. To learn a generalizable multi-task policy with few demonstrations, the pre-training phase of DNAct leverages neural rendering to distill 2D semantic features from foundation models such as Stable Diffusion to a 3D space, which provides a comprehensive semantic understanding regarding the scene. Consequently, it allows various applications to challenging robotic tasks requiring rich 3D semantics and accurate geometry. Furthermore, we introduce a novel approach utilizing diffusion training to learn a vision and language feature that encapsulates the inherent multi-modality in the multi-task demonstrations. By reconstructing the action sequences from different tasks via the diffusion process, the model is capable of distinguishing different modalities and thus improving the robustness and the generalizability of the learned representation. DNAct significantly surpasses SOTA NeRF-based multi-task manipulation approaches with over 30% improvement in success rate. Videos: nerfuser.github.io.

## 1 Introduction

Learning to perform multi-task robotic manipulation in complex environments is a promising direction for future applications, including household robots. Recent studies have showcased the potential of this approach (Shridhar et al., 2023b), demonstrating the possibility of training a single model capable of accomplishing multiple tasks. However, there are still significant issues to overcome. (i) To learn a generalizable multi-task policy from scratch, large-scale datasets are required for comprehensive 3D semantic understanding, which poses a challenge for real-world situations where only a small number of demonstrations are present. (ii) To learn a reliable policy that is capable of executing manipulation in complex environments, the ability to identify the multi-modality of trajectories from the given demonstrations is required. For example, considering the case where the demonstrations contain several possible trajectories avoiding cluttered objects in the kitchen to pick up a knife, a policy learning approach without handling the multi-modality may be biased toward one of the modes (trajectories) and fail to generalize to novel objects or arrangements.

To tackle complex tasks in a more challenging environment (*e.g.*, partial occlusion, various object shapes, and spatial relationship), it is essential to have a comprehensive geometry understanding of the scene. To achieve this, recent work (Driess et al., 2022) reconstructs a 3D scene representation by rendering with Neural Radiance Fields (NeRFs) (Mildenhall et al., 2021). Despite its success in single-task settings like hanging mugs requiring accurate geometry, it requires segmentation of individual objects and lacks semantic understanding of the whole scene, which makes it impractical in more complex environments. On the other hand, to learn the inherent multi-modality in the given demonstrations, previous works (Chi et al., 2023; Wang et al., 2022) adopts diffusion models (Ho et al., 2020; Sohl-Dickstein et al., 2015) to learn an expressive generative policy that possesses the potential to identify different modes, which in turns elevates its success rate. However, in these studies, only relatively simple tasks such as single-task learning are considered. We further observe that in our experiments where the agent is required to perform multi-task manipulation, it is very challenging to leverage diffusion policies to predict accurate action sequences and substantial parameter tuning may be required for more complex tasks. Additionally, the extensive inference time of diffusion models is

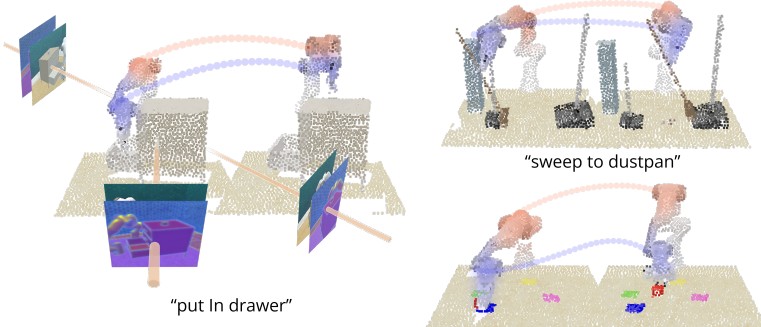

Figure 1: We pre-train a 3D encoder to learn generic semantic features. This is accomplished by reconstructing novel views and corresponding features from Stable Diffusion. The subsequent policy learning is optimized and guided by the diffusion model to leverage multi-modality arising from the multi-task trajectories.

a significant drawback, preventing their application to real robots and complex tasks that require both accuracy and swift inference.

To address these challenges, we learn a unified 3D representation by integrating neural rendering pre-training and diffusion training. Specifically, we first pre-train a 3D encoder via neural rendering with NeRF. We not only utilize the neural rendering to synthesize novel views in RGB but also predict the corresponding semantic features from 2D foundation models. From distilling pre-trained 2D foundation models, we learn a generalizable 3D representation with commonsense priors from internet-scale datasets. We show that this pre-training representation equips the policy with out-of-distribution generalization ability. Differing from previous work (Shim et al., 2023), our method presents no requirement for task-specific in-domain data for 3D pre-training. To optimize the model to distinguish multi-modality from the multi-task demonstrations as suggested in Figure 1, we perform diffusion training to supervise the representation. The diffusion training of DNAct involves the optimization of a feature-conditioned noise predictor and the predictor is designed to reconstruct action sequences across different tasks. With the reconstruction process, the model can effectively identify the inherent multi-modality in the trajectories without biasing toward a certain mode. The learned policy is able to achieve a higher success rate not only in the training environments but also in the environments where novel objects and arrangement is present due to the structured and generalizable representation optimized by the diffusion training. Figure 1 summarizes the concept proposed in DNAct. It illustrates that the 3D semantic feature is learned at the pre-training phase by reconstructing RGB and the corresponding 2D semantic features. In addition, the proposed diffusion training further enhances the ability to discriminate various modes arising from different tasks.

Our contributions are summarized below:

1. We leverage NeRF as a 3D pre-training approach to learn a 3D representation that can unify both semantics and geometry. From distilling pre-trained 2D foundation models into 3D space, we learn a generalizable 3D semantic representation with commonsense priors from internet-scale datasets. We show that this pre-trained representation achieves strong out-of-distribution generalization ability. Different from other NeRF-based methods (Driess et al., 2022; Ze et al., 2023b; Li et al., 2022b), our method eliminates the need for task-specific in-domain datasets consisting of multi-view images paired with action data for 3D pre-training, which can be expensive to collect in the real world.

2. We introduce a novel approach utilizing diffusion training to learn a vision and language feature that distinguishes multi-modality from multi-task demonstrations. By utilizing a feature-conditioned noise predictor and reconstructing the action sequences from different tasks via the diffusion process, the model is capable of distinguishing different modalities and thus improving the robustness and the generalizability of the learned representation. The additionally learned policy network bypasses the need for the time-consuming denoising phase and makes the multi-task learning more stable.

3. DNAct achieves a 1.35x performance improvement in simulation and a 1.33x improvement in real robot experiments while utilizing only 30% of the parameters required by baseline methods. Our experiments also show that the proposed DNAct even outperforms the baseline methods by 1.25x when DNAct is pre-trained with orthogonal tasks that are not used for the following training and evaluation. Additionally, our ablation study on generalization

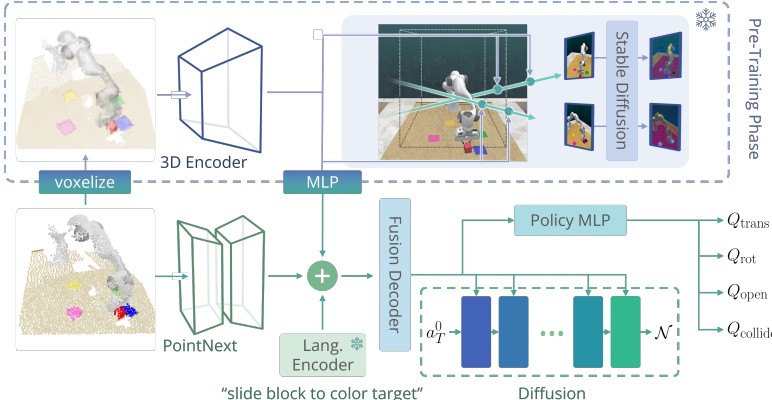

Figure 2: The diagram provides an overview of the proposed DNAct. The upper section of the figure represents the pre-training component, which is frozen during the subsequent training phase, as indicated by the snowflake icon. The area shaded in gray does not participate in this training phase, with only the 3D encoder being utilized to provide generic semantic features. The lower section of the figure corresponds to the training phase, where the diffusion training and the policy MLP are jointly optimized. $a_0^T$ suggests an action sequence of length $T$ and $\mathcal{N}$ is the normal distribution.

demonstrates that DNAct significantly outperforms the baseline by 1.70x when novel objects are present.

## 2    METHOD

In this section, we elaborate on our proposed method, DNAct. The DNAct learns a language-conditioned multi-task policy by leveraging a pre-trained 3D encoder from neural rendering. The 3D encoder is then frozen and paired with a point-cloud encoder, which is trained from scratch. Together, they work to predict representations of observations that preserve the multi-modality within the trajectories, utilizing a diffusion training approach.

### 2.1    ROBOTIC MANIPULATION AS KEYFRAME PREDICTION

Given the computational demands and data inefficiency inherent in continuous action prediction, we reformulate the problem of learning from demonstrations in robotic manipulation as a keyframe-prediction problem (Shridhar et al., 2023a; James et al., 2022). Keyframes are identified from expert demonstrations using a specific metric: a frame is considered a keyframe if the joint velocities approach zero and the gripper maintains a consistent open state. Subsequently, the model is trained to predict the subsequent keyframe based on the current point-cloud observations. This keyframe-prediction problem retains a similarity to the original learning from demonstrations in terms of the training procedure whereas to represent the action of the robot arm with a gripper, we estimate the translation $a_{\text{trans}} \in \mathbf{R}^3$, rotation $a_{\text{rot}} \in \mathbf{R}^{(360/5) \times 3}$, gripper openness $a_{\text{open}} \in [0, 1]$, and collision avoidance $a_{\text{collision}} \in [0, 1]$. The rotation action is discretized into 72 bins for 5 degrees each bin per rotation axis, and the collision avoidance indicator $a_{\text{collision}}$ estimates the need for contact. This approach effectively transforms the computationally intensive continuous control problem into a more manageable, discretized keyframe-prediction task, allowing the motion planner to handle the intricate procedures.

### 2.2    3D ENCODER PRE-TRAINING WITH NEURAL FEATURE FIELDS

To develop a scene-agnostic 3D encoder capable of providing comprehensive 3D semantic information of complex environments, we leverage neural rendering and knowledge distillation for 3D encoder pre-training. This method constructs 3D semantics from a 2D foundational model through neural rendering with NeRF. With higher sample efficiency, this pre-training paradigm significantly facilitates training efficiency and model performance across various task domains.

When learning from a limited set of demonstrations, our method of pre-training and subsequently using the frozen 3D encoder for downstream tasks ensures that the subsequent decision-making process leverages accurate 3D semantics rather than overfitting to the demonstrations with behavior cloning. Additionally, the pre-training does not necessitate paired action in the training dataset, expanding the possibility of utilizing a broader range of training datasets unrelated to the target tasks

and scenes directly. This approach eliminates the need for task-specific data in both NeRF and policy training, enhancing scalability and versatility.

Firstly, we convert point-cloud observations into a $100^3$ voxel. Our 3D encoder then encodes the voxel into a volume feature, denoted as $v \in \mathbb{R}^{100^3 \times 64}$. We then sample point feature $v_{\mathbf{x}} \in \mathbb{R}^{64}$ from the volume feature by employing a trilinear interpolation operation to approximate $v_{\mathbf{x}}$ over a discrete grid in 3D space. Three distinct functions are further utilized for volumetric rendering of novel views and corresponding vision-language features:

1. A density function $\sigma(\mathbf{x}, v_{\mathbf{x}}) : \mathbb{R}^{3+64} \mapsto \mathbb{R}_+$, mapping the 3D point $\mathbf{x}$ and the 3D feature $v_{\mathbf{x}}$ to density $\sigma$.

2. An RGB function $\mathbf{c}(\mathbf{x}, \mathbf{d}, v_{\mathbf{x}}) : \mathbb{R}^{3+3+64} \mapsto \mathbb{R}^3$, assigning color to the 3D point $\mathbf{x}$, the view direction $\mathbf{d}$, and the 3D feature $v_{\mathbf{x}}$.

3. A vision-language embedding function $\mathbf{f}(\mathbf{x}, \mathbf{d}, v_{\mathbf{x}}) : \mathbb{R}^{3+3+64} \mapsto \mathbb{R}^{512}$, correlating the 3D point $\mathbf{x}$, the view direction $\mathbf{d}$, and the 3D feature $v_{\mathbf{x}}$ to the vision-language embedding.

The rendered color and feature embedding are estimated through integration along the ray:

$$\hat{\mathbf{C}}(\mathbf{r}, v) = \int_{t_n}^{t_f} T(t)\sigma(\mathbf{r}(t), v_{\mathbf{x}(t)})\mathbf{c}(\mathbf{r}(t), \mathbf{d}, v_{\mathbf{x}(t)})\mathrm{d}t\,,$$
$$\hat{\mathbf{F}}(\mathbf{r}, v) = \int_{t_n}^{t_f} T(t)\sigma(\mathbf{r}(t), v_{\mathbf{x}(t)})\mathbf{f}(\mathbf{r}(t), \mathbf{d}, v_{\mathbf{x}(t)})\mathrm{d}t\,,$$

$(1)$

where $T(t) = \exp\left(-\int_{t_n}^{t} \sigma(s)\mathrm{d}s\right)$, $\mathbf{r}(t) = \mathbf{o} + t\mathbf{d}$ represents the camera ray, $\mathbf{o} \in \mathbb{R}^3$ is the camera's origin coordinate, $\mathbf{d}$ is the view direction, and $t$ is depth, bounded by $[t_n, t_f]$.

Our 3D encoder is optimized by reconstructing the multi-view RGB images and vision language embedding through the following MSE loss:

$$\mathcal{L}_{\text{recon}} = \sum_{\mathbf{r} \in \mathcal{R}} \|\mathbf{C}(\mathbf{r}) - \hat{\mathbf{C}}(\mathbf{r})\|_2^2 + \lambda_{\text{feat}}\|\mathbf{F}(\mathbf{r}) - \hat{\mathbf{F}}(\mathbf{r})\|_2^2\,,$$

$(2)$

where $\mathbf{C}(\mathbf{r})$ is the ground truth color, $\mathbf{F}(\mathbf{r})$ is the ground truth embedding generated by a vision-language foundation model. For more information about $\mathbf{F}(\mathbf{r})$, please refer to the Stable Diffusion Feature Extraction section in the experimental setup. $\mathcal{R}$ represents the set of rays emanating from camera poses, and $\lambda_{\text{feat}}$ is the weight attributed to the embedding reconstruction loss. Lastly, similar to the manner in which we approximate the 3D feature $v_{\mathbf{x}}$, we utilize trilinear interpolation on the features predicted by the 3D encoder to derive the features corresponding to points in the point cloud.

## 2.3 DIFFUSION-GUIDED FEATURE LEARNING WITH ACTION SEQUENCES

For the subsequent training phase, to facilitate the agent's utilization of pre-trained 3D semantic features and to ensure adaptability to a variety of tasks, the downstream action prediction models additionally incorporate a point-cloud feature, encoded by a PointNext network (Qian et al., 2022). Specifically, given the pointcloud observation, we sample 4096 points as input of the PointNext to learn a geometric point features, denoted as $v_p \in \mathbb{R}^{4096 \times 32}$. The sampled points are also proceeded to query the 3D semantic features from pre-trained 3D volume feature $v$, denoted as $v_s \in \mathbb{R}^{4096 \times 64}$. The robot's proprioception is projected into a 32-dimensional vector by applying a linear layer, while the language goal features from CLIP are projected into a 64-dimensional vector. Both vectors are then attached to each sampled point, resulting in the sequence of size $4096 \times 32$ and $4096 \times 64$ for the robot's proprioception and language respectively. We concatenate all features together, including pre-trained semantic features $v_s$, point-cloud features $v_p$, robot proprioception, and language embeddings. This combined feature is further fused through several set abstraction layers (Qi et al., 2017) to obtain the final vision-language embedding $v_f$.

Through our multi-task robotic manipulation experiments, we demonstrate that learning with both the pre-trained features and the newly encoded features from PointNext offers notable generalization and an enhanced capability to capture semantic information. This underscores the effectiveness of a multi-task manipulation agent when combining insights derived from a pre-trained 3D encoder with those from a learning-from-scratch encoder.

### 2.3.1 DIFFUSION MODELS AS REPRESENTATION LEARNING

To further effectively integrate pre-trained semantic features $v_s$, point-cloud features $v_p$, and language embeddings $v_l$ from foundation models like CLIP, we formulate the representation learning as an action sequence reconstruction problem with Denoising Diffusion Probabilistic Models (DDPMs) (Ho et al., 2020). Learning representation with diffusion models encapsulates the multi-modality inherent in multi-task demonstrations into a fused representation.

Starting from a random action sequence $a_T^K$ of length $T$ sampled from Gaussian noise, the DDPM performs $K$ iterations of denoising to produce $a_T^k, a_T^{k-1}, ..., a_T^0$, until the action reconstruction is achieved. We perform the action denoising process with a fused feature-conditioned noise predictor via Feature-wise Linear Modulation (Perez et al., 2018).

For each training instance, we randomly select a denoising iteration $k$ and sample a random noise $\epsilon^k$ with a variance schedule. The conditional noise prediction network is optimized to predict the noise from the data sample with noise added:

$$\mathcal{L}_{\text{diff}} = \mathbb{E}\|\epsilon^k - \epsilon_\theta(v_f, a_T^0 + \epsilon, k)\|_2^2, \tag{3}$$

where $v_f$ is the output of the fusion decoder (Fig. 2) with concatenated $v_s, v_p, v_l$ as inputs. The gradient from the loss function propagates through the noise decoder, the fusion decoder, and lastly the PointNext architecture. As shown in Ho et al. (2020), minimizing the loss function (Eq. 3) essentially minimizes the variational lower bound of the KL-divergence between the data distribution and the distribution of samples drawn from the DDPM.

To predict actions for robotic manipulation, we optimize a policy network that estimates the Q-values for each action: $Q_{\text{trans}}, Q_{\text{rot}}, Q_{\text{open}},$ and $Q_{\text{collision}}$ corresponding to $a_{\text{trans}}, a_{\text{rot}}, a_{\text{open}},$ and $a_{\text{collision}}$ via behavior cloning. This policy network shares the fused feature, $v_f$, with the diffusion model. The objective for DNAct at the training phase can be summarized as:

$$\mathcal{L} = \mathcal{L}_{\text{bc}} + \lambda_{\text{diff}}\mathcal{L}_{\text{diff}}, \tag{4}$$

where

$$\mathcal{L}_{\text{bc}} = -\lambda_{\text{trans}}|Q_{\text{trans}} - Y_{\text{trans}}| - \lambda_{\text{rot}}\mathbb{E}_{Y_{\text{rot}}}\left[\log \mathcal{V}_{\text{rot}}\right] - \lambda_{\text{open}}\mathbb{E}_{Y_{\text{open}}}\left[\log \mathcal{V}_{\text{open}}\right] - \lambda_{\text{collide}}\mathbb{E}_{Y_{\text{collide}}}\left[\log \mathcal{V}_{\text{collide}}\right] \tag{5}$$

$\mathcal{V}_i = \text{softmax}(\mathcal{Q}_i)$ for $\mathcal{Q}_i \in [\mathcal{Q}_{\text{open}}, \mathcal{Q}_{\text{rot}}, \mathcal{Q}_{\text{collide}}]$ and $Y_i \in [Y_{\text{rot}}, Y_{\text{open}}, Y_{\text{collide}}]$ is the ground truth one-hot encoding. For translation, we calculate $L1$ loss between a predicted continuous 3D coordinate $\mathcal{Q}_{\text{trans}}$ and the ground truth $Y_{\text{trans}}$.

While using a diffusion model for action prediction with the denoised $a_T^0$ has been recognized for its potential to achieve state-of-the-art performance, it is empirically observed to be highly sensitive to specific architectures and configurations. For example, a balance is needed in the network architecture's expressiveness to avoid unstable training while ensuring capability (Hansen-Estruch et al., 2023). Additionally, varying configurations and hyper-parameters are required for different tasks (Chi et al., 2023), complicating the learning process for a single multi-task agent.

Consequently, we propose to learn the multi-modalities inherent in multi-task demonstrations with diffusion training. Rather than deploying the generated actions from the diffusion model for robot control during inference, we employ an additional policy network optimized jointly to predict actions using the fused feature shared with the diffusion model. Our method, with the additional policy network, has two key benefits: (i) It ensures quicker action inference as the policy network predicts actions faster than the diffusion model, which requires multiple denoising steps. (ii) The policy network compensates for the diffusion model's limitations in action prediction accuracy, which enhances the robustness of the entire training pipeline regarding hyper-parameters and configurations.

## 3 EXPERIMENTS

### 3.1 EXPERIMENT SETUP

**Simulation.** We conduct all simulated experiments on RLBench (James et al., 2020), a challenging large-scale benchmark and learning environment for robot learning. A 7 DoF Franka Emika Panda

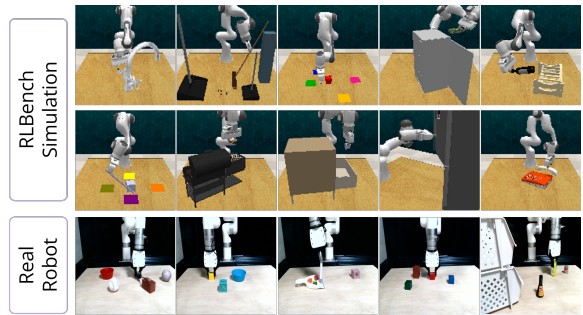

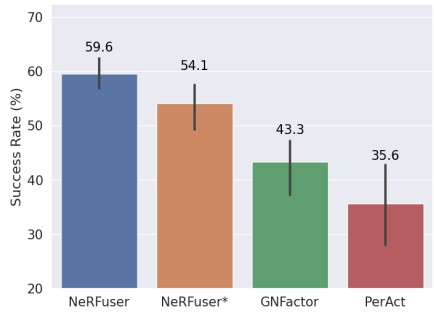

Figure 3: The ten RLBench and five real robot tasks in our experiments.

Figure 4: Average success rates across 3 seeds on RLBench. The error bar shows one standard deviation.

robot arm is equipped with a parallel gripper and affixed to a table for the execution of various tasks. Visual observations are collected from four RGB-D cameras, placed at the front, left shoulder, right shoulder, and wrist with a resolution of 128×128. We convert obtained RGB and depth images to point cloud using camera intrinsics. Each task has multiple variations on different aspects like shape and color with textual summarizing descriptions. We select 10 challenging language-conditioned tasks with 50 demonstrations per task for training. To succeed with limited training, the agent needs to learn generalizable manipulation skills instead of just overfitting to the training examples.

**Real Robot.** We set up a tabletop environment using the xArm7 robot with a parallel gripper for real robot experiments. We design 5 tasks with distractors in the scene, including *Put in Bowel*, *Stack Blocks*, *Hit Ball*, *Put in Bin*, *Sweep to Dustpan*, as shown in the Figure 3. We place three RealSense cameras around the robot to capture multi-view images for Neural Radiance Fields training. For policy training and evaluation, we only use the front RealSense camera to obtain RGB-D images and convert them to point cloud input. Example real-robot tasks are shown in Figure 7.

**Stable Diffusion Feature Extraction.** We use RGB images and corresponding language instructions as input for the Stable Diffusion model. Specifically, we encode the language instruction with a pre-trained text encoder and extract the text-to-image diffusion UNet's internal features by feeding language embedding and RGB images into the UNet without adding noise. The rich and dense diffusion features are then extracted from the last layer of the UNet blocks. We only require a single forward pass to extract visual features instead of going through the entire multi-step generative diffusion process.

**Demonstration Collection.** We collect 50 demonstrations for each task in the RLBench environment using RRT-Connect (Kuffner & LaValle, 2000) motion planner. For real-world experiments, we collect 10 demonstrations using Linear Motion for each task with an HTC Vive controller.

**Training Details.** We train DNAct for 50K iterations with a batch size of 32 with two NVIDIA RTX3090 GPUs, taking less than 12 hours. We train PerAct and GNFactor for 200K iterations with a batch size of 2 taking over 2 and 4 days, respectively.

## 3.2 SIMULATION RESULTS

**Multi-Task Performance.** As shown in Table 1 and 2, DNAct outperforms PerAct and GNFactor across various tasks, especially in challenging long-horizon and fine-grained tasks. For example, in *open fridge* task, it requires the agent to have accurate 3D geometry understanding to precisely grasp the door and open it without collisions. DNAct can achieve a success rate of 22.7% while both PerAct and GNFactor demonstrate a minor success rate on it. For *put in drawer* task, the robot needs to first open the drawer, pick up the item, and finally put it in the drawer. It presents a comprehensive challenge to robots in long-horizon planning. We find that DNAct achieves a 54.7% success rate, largely outperforming PerAct and GNFactor. Notably, DNAct only uses 11.1M parameters, significantly less than PerAct (33.2M parameters) and GNFactor (41.7M parameters). These results demonstrate that DNAct is both more efficient and capable of handling various tasks.

**Generalization Performance to Unseen Tasks.** The ability of a robotic agent to generalize to unseen environments with novel attributes is critical for operational versatility and adaptability. Such environments often introduce challenges, including distracting objects, larger objects, and new positional contexts. As illustrated in Table 3, our proposed method, DNAct, showcases robustness and adaptability, achieving significantly higher performance compared to the baseline methods, PerAct and GNFactor.

Table 1: **Multi-Task Performance on RLBench.** We evaluate 25 episodes for each checkpoint on 10 tasks across 3 seeds and report the success rates (%) of the final checkpoints. Our method outperforms the most competitive baseline PerAct (Shridhar et al., 2023a) and GNFactor (Ze et al., 2023b) with an average improvement of **1.67**x and **1.38**x

| Method / Task | turn tap | drag stick | open fridge | put in drawer | sweep to dustpan | **Average** |
|---|---|---|---|---|---|---|
| PerAct | $57.3_{\pm6.1}$ | $14.7_{\pm6.1}$ | $4.0_{\pm6.9}$ | $16.0_{\pm13.9}$ | $4.0_{\pm6.9}$ | |
| GNFactor | $56.0_{\pm14.4}$ | $68.0_{\pm38.6}$ | $2.7_{\pm4.6}$ | $12.0_{\pm6.9}$ | $61.3_{\pm6.1}$ | |
| DNAct | $73.3_{\pm4.6}$ | $97.3_{\pm4.6}$ | $22.7_{\pm16.2}$ | $54.7_{\pm22.0}$ | $24.0_{\pm17.4}$ | |

| Method / Task | meat off grill | phone on base | place wine | slide block | put in safe | |
|---|---|---|---|---|---|---|
| PerAct | $77.3_{\pm14.0}$ | $98.7_{\pm2.3}$ | $6.7_{\pm6.1}$ | $29.3_{\pm22.0}$ | $48.0_{\pm26.2}$ | 35.6 |
| GNFactor | $77.3_{\pm9.2}$ | $96.0_{\pm6.9}$ | $8.0_{\pm6.9}$ | $21.3_{\pm6.1}$ | $30.7_{\pm6.1}$ | 43.3 |
| DNAct | $92.0_{\pm6.9}$ | $82.7_{\pm12.2}$ | $65.3_{\pm9.2}$ | $58.7_{\pm20.5}$ | $25.3_{\pm8.3}$ | **59.6** |

Table 2: We also report the success rates (%) of the **best checkpoints** for each tasks across 3 seeds. Our method achieves **1.65**x and **1.30**x improvement compared with PerAct and GNFactor

| Method / Task | turn tap | drag stick | open fridge | put in drawer | sweep to dustpan | **Average** |
|---|---|---|---|---|---|---|
| PerAct | $66.7_{\pm4.6}$ | $38.7_{\pm6.1}$ | $12.0_{\pm6.9}$ | $30.7_{\pm6.1}$ | $4.0_{\pm6.9}$ | |
| GNFactor | $73.3_{\pm6.1}$ | $92.0_{\pm8.0}$ | $17.3_{\pm6.1}$ | $26.7_{\pm12.2}$ | $68.0_{\pm8.0}$ | |
| DNAct | $85.3_{\pm6.1}$ | $100.0_{\pm0.0}$ | $28.0_{\pm10.6}$ | $70.7_{\pm12.9}$ | $86.7_{\pm2.3}$ | |

| Method / Task | meat off grill | phone on base | place wine | slide block | put in safe | |
|---|---|---|---|---|---|---|
| PerAct | $85.3_{\pm8.3}$ | $100.0_{\pm0.0}$ | $13.3_{\pm2.3}$ | $38.7_{\pm6.1}$ | $62.7_{\pm11.5}$ | 45.2 |
| GNFactor | $81.3_{\pm8.3}$ | $100.0_{\pm0.0}$ | $29.3_{\pm2.3}$ | $42.7_{\pm12.2}$ | $45.3_{\pm4.6}$ | 57.6 |
| DNAct | $96.0_{\pm4.0}$ | $89.3_{\pm6.1}$ | $72.0_{\pm6.9}$ | $84.0_{\pm8.0}$ | $34.7_{\pm6.1}$ | **74.7** |

This superior performance of DNAct is attributed to the 3D semantic pre-training and diffusion training. These practices allow DNAct to optimize the learned representation from different perspectives, such as knowledge distillation and action sequence reconstruction, going beyond merely optimizing the behavior cloning objective. This prevents our model from overfitting to the training demonstrations on predicting actions.

DNAct is particularly proficient in situations impacted by external distractors, size alterations of objects, and novel positions, maintaining high performance amidst diverse challenges. This underscores DNAct's superior generalization capabilities and sophisticated learning mechanism, marking it as a promising advancement in the domain of robotic manipulation.

**Pre-Training on Out-of-Domain Data.** Typically, NeRF-based imitation/reinforcement learning requires task-specific data for both neural rendering and policy learning. However, in the proposed DNAct, employing pre-trained representations can enhance scalability with out-of-domain data. We delve deeper into the efficacy of DNAct by utilizing five out-of-domain tasks, each with fifty demonstrations, to pre-train the 3D encoder via neural rendering; this variant is denoted as DNAct*. As depicted in Figure 4, DNAct* attains a substantially higher success rate compared to both PerAct and GNFactor and showcases performance merely slightly worse than DNAct, which utilizes in-domain data. This underscores the significant potential of leveraging out-of-domain data for pre-training a 3D encoder through neural rendering. We hypothesize that pre-training with even larger-scale out-of-domain datasets could yield more pronounced effectiveness. We also show the visualization of the 3D point feature extracted from the pre-trained 3D encoder as in Figure 6.

**Ablation.** We conduct ablation experiments to analyze the impact of several key components in DNAct. From the following experiment, we obtained several insights:

(i) Pre-trained representations via neural rendering are crucial in multi-task robot learning. As shown in table 4, a Learning-from-Scratch PointNext denoted as LfS, has a much lower success rate without fusing pre-trained 3D representations. This strongly demonstrates the efficacy of pre-training 3D representations by distilling the foundation model feature.

(ii) The ablation study reveals a 13.09% performance decrease when not learning jointly with diffusion training. This suggests that the representations optimized by the diffusion model enable the policy network to make more comprehensive decisions compared to solely optimizing the behavior cloning objective. Beyond the inherent benefits of diffusion training, the optimization of the diffusion model with action sequences potentially embeds action sequence information into the representation. This enhancement is crucial, improving the policy network, which predicts only one step at each timestep.

(iii) We further investigate the impact of different foundation models for feature distillation. By replacing the stable diffusion model with the foundation model DINOv2 (Oquab et al., 2023) and vison-language foundation model CLIP (Radford et al., 2021), both models have slightly lower success rates compared with the stable diffusion model. This may be because SD features provide a higher quality spatial information. However, DNAct with these two features still outperforms the baseline by a large margin, demonstrating the remarkable ability of foundation models.

Table 3: **Generalization to unseen tasks on RLBench.** We evaluate 25 episodes for each task with the final checkpoint across 3 seeds. We denote "N" as a new position, "L" as a larger object and "D" as adding a distractor.

| Method / Task | drawer (D) | place (N) | slide (L) | meat (N) | safe (D) | Average |
|---|---|---|---|---|---|---|
| PerAct | $5.3_{\pm 2.3}$ | $2.7_{\pm 4.6}$ | $32.0_{\pm 14.4}$ | $66.7_{\pm 23.1}$ | $42.7_{\pm 15.1}$ | 29.8 |
| GNFactor | $18.7_{\pm 6.1}$ | $12.0_{\pm 10.6}$ | $22.7_{\pm 18.0}$ | $72.0_{\pm 10.6}$ | $29.3_{\pm 23.1}$ | 30.9 |
| DNAct | $\mathbf{52.0}_{\pm \mathbf{8.0}}$ | $\mathbf{30.7}_{\pm \mathbf{16.2}}$ | $\mathbf{54.7}_{\pm \mathbf{28.1}}$ | $\mathbf{93.3}_{\pm \mathbf{2.3}}$ | $30.7_{\pm 22.0}$ | **52.3** |

Table 4: An ablation study on different components of DNAct. LfS indicates learning from scratch and Task-agnostic Pre-training suggests that a dataset from a different set of tasks are used for the pre-training phase.

| Ablation | Success Rate (%) |
|---|---|
| DNAct | **59.6** |
| LfS | 33.2 |
| w/o. Diffusion objective | 51.8 |
| Stable Diffusion → DINOv2 | 52.4 |
| Stable Diffusion → CLIP | 53.2 |
| Task-agnostic Pre-training | 54.1 |

Table 5: Success rates (%) on real robot tasks.

| Tasks | PerAct | DNAct |
|---|---|---|
| Put in Bowel | 40 | 50 |
| Stack Blocks | 40 | 30 |
| Hit Ball | 30 | 60 |
| Put in Bin | 60 | 80 |
| Sweep to Dustpan | 40 | 60 |
| Average | 42 | **56** |

## 3.3 REAL-ROBOT RESULTS

We summarize the performance of DNAct on 5 real robot tasks in table 5. As shown in Figure 5, we provide the novel view synthesis results of DNAct on both RGB and diffusion features in the real world. It shows DNAct parses all object instances and extracts scene semantics effectively. From the experiments, DNAct achieves an impressive average success rate of 56%, while PerAct attains 42%. Notably, in the task *Hit ball*, it requires the robot to pick the screwdriver, locate the ball, and hit it with the presence of multiple distractors. DNAct achieves a 60% success rate in this challenging task, largely outperforming the 30% success rate of PerAct. Another example is the *Sweep to Dustpan* task, where the robot arm needs to identify the location of the debris and the dustpan, and precisely grasp the broom with complex motion. Although DNAct demonstrates strong capability, it still lacks the ability to deal with scenes characterized by intricate spatial relationships and novel language commands. This challenge could potentially be addressed through the integration of a multi-modal large language model, leveraging its zero-shot generalization capabilities and enhanced reasoning skills to augment DNAct' performance.

## 4 RELATED WORK

**Multi-Task Robotic Manipulation** Recent advancements in multi-task robotic manipulation are making significant improvements in performing complex tasks and adapting to new scenarios (Rahmatizadeh et al., 2018; Jang et al., 2022; Brohan et al., 2022; Shridhar et al., 2022; Yu et al., 2020; Yang et al., 2020). Many standout methods generally use extensive interaction data to develop multi-task models (Jang et al., 2022; Brohan et al., 2022; Shridhar et al., 2022; Kalashnikov et al., 2018). For instance, RT-1 and RT-2 (Brohan et al., 2022; 2023) have demonstrated improved performance in real-world robotic tasks across various datasets.

To mitigate the need for extensive demonstrations, techniques that use keyframes have been shown effective (Song et al., 2020; Murali et al., 2020; Mousavian et al., 2019; Xu et al., 2022; Li et al., 2022a; Ze et al., 2023b). PerAct (Shridhar et al., 2023a) incorporates the Perceiver Transformer (Jaegle et al., 2021) to understand language goals and voxel observations, proving its value in real-world robot experiments. In our study, we are using the same action prediction model as PerAct, but we are focusing on increasing the model's ability to generalize by learning a generalized and multimodal representation with limited data.

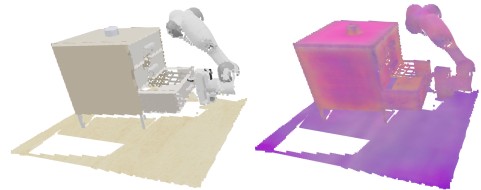

Point Cloud     3D Point Feature

Figure 6: **3D point features.** The visualization shows 3D point features extracted from the pretrained 3D encoder using 5 unseen tasks. We show that our model can effectively learn a 3D semantic representation by utilizing out-of-domain data.

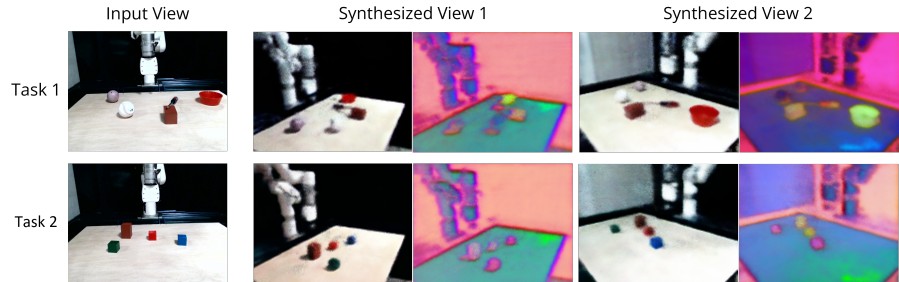

Figure 5: **Multi-View synthesis of DNAct in the real world.** We learn 3D semantic representations by distilling vision foundation models. The rendered feature maps are visualized on the right side.

**3D Representations for Policy Learning.**

To enhance manipulation policies using visual information, many studies have focused on improving 3D representations. Ze et al. (2023a) employ a 3D autoencoder, showing improved sample efficiency in motor control compared to 2D methods. Driess et al. (2022) uses NeRF for learning state representation, demonstrating initial success but with limitations, such as the requirement for object masks and the absence of scene structure and the robot arm, affecting its real-world applicability.

GNFactor (Ze et al., 2023b), similar to our proposed DNAct, utilize a vision foundation model in NeRF. However, GNFactor, by jointly optimizing NeRF and behavior cloning, reduces the quality of neural rendering and limits the benefits from the semantic information. Our DNAct addresses these challenges by learning with scene-agnostic semantic and adaptive point-cloud features in multi-task real-world scenarios, showcasing potential for real applications with a smaller number of parameters and faster inference time.

**Neural Radiance Fields.** Over the years, neural fields have made significant advancements in novel view synthesis and scene representation learning (Chen et al., 2021; Mescheder et al., 2019; Mildenhall et al., 2021; Niemeyer et al., 2019; Park et al., 2019; Sitzmann et al., 2019), with efforts to merge them into robotics (Jiang et al., 2021; Lin et al., 2023b; Li et al., 2022b; Driess et al., 2022; Shim et al., 2023). NeRF (Mildenhall et al., 2021) and its variants (Chen & Wang, 2022; Jang & Agapito, 2021; Lin et al., 2023a; Reizenstein et al., 2021; Rematas et al., 2021; Trevithick & Yang, 2021; Wang et al., 2021) have been remarkable for enabling photorealistic view synthesis by learning a scene's implicit function, but its need for per-scene optimization limits its generalization.

**Diffusion Models for Policy Learning.** The growing interest in diffusion models (Sohl-Dickstein et al., 2015; Ho et al., 2020) has led to important advancements in RL and robotic learning methods. This interest is due to the success of generative modeling in tackling various decision-making problems, with applications in both policy and robotic learning domains (Janner et al., 2022; Liang et al., 2023; Wang et al., 2022; Argenson & Dulac-Arnold, 2020; Chen et al., 2022; Ajay et al., 2022; Urain et al., 2022; Huang et al., 2023; Chi et al., 2023). In robotics, Janner et al. (2022) and Huang et al. (2023) have explored the use of diffusion models in planning to infer possible action trajectories in different environments. This research aligns with works like Hansen-Estruch et al. (2023) reinterpreting Implicit Q-learning using diffusion parameterized behavior policies. Unlike these methods, the proposed DNAct doesn't directly use diffusion models for action inference but uses them to derive multi-modal representation and jointly optimize a policy network, offering more stable training and eliminating the need for the costly inference of diffusion models.

## 5 CONCLUSION

In this work, we propose DNAct, a generalized multi-task policy learning approach for robotic manipulation. DNAct leverages a NeRF pre-training to distill 2D semantic features from foundation models to a 3D space, which provides a comprehensive semantic understanding regarding the scene. To further learn a vision and language feature that encapsulates the inherent multi-modality in the multi-task demonstrations, DNAct is optimized to reconstructs the action sequences from different tasks via diffusion process. The proposed diffusion training at the training phase enables the model to not only distinguish representation of various modalities arising from multi-task demonstrations but also become more robust and generalizable for novel objects and arrangements. DNAct shows promising results in both simulation and real world experiments. It outperforms baseline approaches by over 30% improvement in 10 RLBench tasks and 5 real robot tasks, illustrating the generalizability of the proposed DNAct. **We are committed to releasing the code.**

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

## A EXPERIMENT DETAILS

To enhance the reproducibility of DNAct, we summarize the experiment details such as the configurations and parameters used in Table 6.

Table 6: Hper-parameters. We list all hyper-parameters for reproducing DNAct results below.

| Parameter | Value |
|---|---|
| training iteration | 50K |
| Learning rate Scheduler | False |
| Optimizer | Adam |
| learning rate | 0.0005 |
| Weight Decay | $1 \times 10^{-6}$ |
| Translation loss coefficient ($\lambda_{\text{trans}}$) | 300 |
| Rotation loss coefficient ($\lambda_{\text{trans}}$) | 1 |
| Gripper openness loss coefficient ($\lambda_{\text{trans}}$) | 1 |
| Collision loss coefficient ($\lambda_{\text{trans}}$) | 1 |
| Diffusion loss coefficient ($\lambda_{\text{diff}}$) | 5 |
| denoising steps (training) | 100 |
| ray batch size $b_{\text{ray}}$ | 512 |
| embedding loss coefficient $\lambda_{\text{feat}}$ | 0.01 |
| Batch size | 32 |
| GPU | RTX 3090 |

## B SIMULATED TASK DESCRIPTIONS

We have chosen ten language-conditioned tasks from RLBench (James et al., 2020). An overview of these tasks is presented in Table 7. Our variations include randomly sampled colors, sizes, placements, and object categories. The color set includes twenty instances: red, maroon, lime, green, blue, navy, yellow, cyan, magenta, silver, gray, orange, olive, purple, teal, azure, violet, rose, black, and white. The size set comprises two types: short and tall. The placements and object categories are specific to each task. The average keyframes numbers are varied from 2 to 15, representing different horizon length.

Table 7: **Language-conditioned tasks in RLBench (James et al., 2020).**

| Task | Variation Type | # of Variations | Avg. Keyframes | Language Template |
|---|---|---|---|---|
| turn tap | placement | 2 | 2.0 | "turn — tap" |
| drag stick | color | 20 | 6.0 | "use the stick to drag the cube onto the — — target" |
| open fridge | placement | 1 | 4.4 | "open the fridge door" |
| put in drawer | placement | 3 | 15.0 | "put the item in the — drawer" |
| sweep to dustpan | size | 2 | 4.6 | "sweep dirt to the — dustpan" |
| meat off grill | category | 2 | 5.0 | "take the — off the grill" |
| phone on base | placement | 1 | 6.4 | "put the phone on the base" |
| place wine | placement | 3 | 6.2 | "stack the wine bottle to the — of the rack" |
| slide block | color | 4 | 4.7 | "slide the block to — target" |
| put in safe | placement | 3 | 6.1 | "put the money away in the safe on the — shelf" |

## C REAL ROBOT KEYFRAMES

We demonstrate the keyframes from two of our real robot tasks.

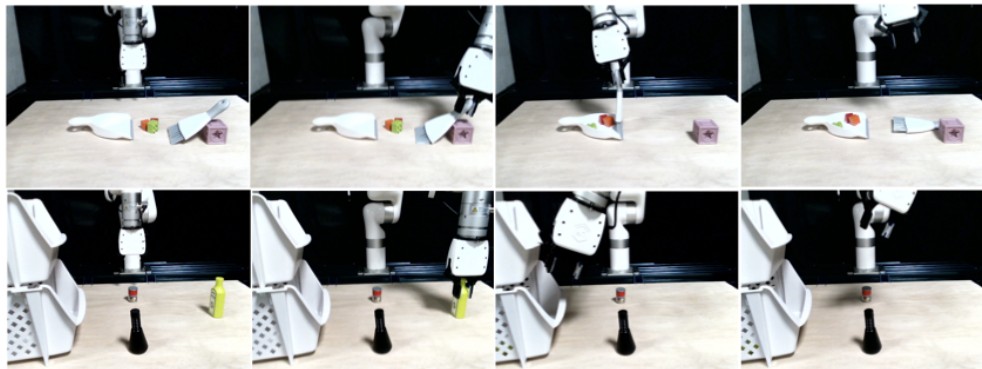

Figure 7: **Keyframes for Real-Robot tasks** We give two examples of keyframes used in our real robot tasks.

## D   MODEL ARCHITECTURES

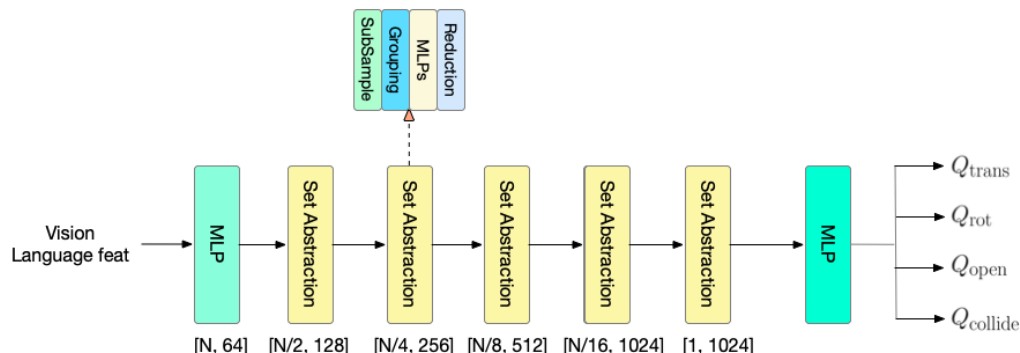

Figure 8: **Fusion Decoder Architecture.**

**3D Encoder.** We use a 3D UNet with $4.72M$ parameters to encode the input voxel $100^3 \times 10$ into a deep 3D volumetric representation of size $100^3 \times 64$. We provide the PyTorch-Style pseudo-code for the forward process as follows. Each conv layer comprises a single 3D Convolutional Layer, followed by Batch Normalization, and Leaky ReLU activation.

```
def forward(self, x):
    conv0 = self.conv0(x) # 100^3x8
    conv2 = self.conv2(self.conv1(conv0)) # 50^3x16
    conv4 = self.conv4(self.conv3(conv2)) # 25^3x32
    conv6 = self.conv6(self.conv5(conv4)) # 13^3x64
    conv8=self.conv8(self.conv7(conv6)) # 7^3x128
    x = self.conv10(self.conv9(conv8)) # 7^3x256
    x = conv8 + self.conv11(x) # 7^3x128
    x = conv6 + self.conv13(x) # 13^3x64
    x = conv4 + self.conv15(x) # 25^3x32
    x = conv2 + self.conv17(x) # 50^3x16
    x = self.conv_out(conv0 + self.conv19(x)) # 100^3x64
    return x
```

**Fusion Decoder.** We use several set abstraction blocks to fuse pre-trained 3D semantics feature, geometric point cloud feature, language feature from CLIP and robot proprioception embedding. This generates a vision-language feature of size $1024$ as input of policy MLP.

**Noise Predictor.** The architecture of the noise predictor is a modified U-Net architecture designed to handle 1D inputs and incorporates conditional inputs. It's comprised of an Encoder, Decoder, and additional components to process conditional inputs.

- **Encoder:** The encoder is composed of a sequence of conditional residual block and each block has two "Conv1d → GroupNorm → Mish" components. A downsampling layer is applied after each block, performing downsampling with Conv1d.

- **Decoder:** Similar to the encoder while replacing the downsampling layers with upsampling layers performed by ConvTranspose1d.

- **Final Convolution Layer:** A sequence comprising a "Conv1d → GroupNorm → Mish" and Conv1d layer.

We apply the Feature-wise Linear Modulation (FiLM) (Perez et al., 2018) to enable the noise predictor to predict the noise with conditional input, the fused feature. This technique is particularly useful in conditional generation tasks. The FiLM module performs modulation by applying a simple affine transformation to each feature map. Given a feature map $x$, the FiLM transformation is defined as:

$$\text{FiLM}(x) = \gamma \cdot x + \beta, \tag{6}$$

where $x$ is the input feature map, $\gamma$ is the scale parameter, and $\beta$ is the shift parameter. In our noise predictor architecture, the fused feature is used to predict the FiLM parameters $\gamma$ and $\beta$. The predicted parameters are then used to modulate the feature maps within each block.

**Policy MLP.** The Policy MLP is composed of several MLPs. The translation output has one independent MLP and the remaining rotation, collision, and open action, share another set of MLP.

