# OpenReview forum: "NeRFuser: Diffusion Guided Multi-Task 3D Policy Learning"
_ICLR.cc/2024/Conference — Submitted to ICLR 2024_

### Official Review · Reviewer_Pcik · 2023-10-12

**Soundness:** 4 excellent
**Presentation:** 4 excellent
**Contribution:** 3 good
**Rating:** 8
**Confidence:** 4

**Summary:**

The paper introduces a method for robot instruction following. In the first phase, it trains a 3D encoder that converts a voxelized representation of a 3D scene to its 64-dimensional latent representation. the encoder is trained such that the latent representation can reconstruct the scene's radiance field as well as features distilled from 2D vision models such as Stable Diffusion, DINO or CLIP. In the second phase, the 3D encoder is frozen, and PointNext + a policy model is trained. the 3D latent representation is used in combination with PointNext features and a pre-trained language encoder to produce fused features, which are then the input to the policy model. The policy model is an MLP trained to predict Q values, alongside a diffusion model that seeks to generate action sequences. The authors demonstrate this method on a number of robotic manipulation tasks.

**Strengths:**

The work is very clearly presented with a nice project page and good figures. The authors will release code.
They perform solid experiments with thorough baseline comparisons and ablations. They make a number of interesting insights, including the superiority of Stable Diffusion as a 2D teacher to CLIP or DINO features, and the ability to pretrain the 3D encoder on out of distribution data and still achieve good performance. The experimental results and model capabilities are not groundbreaking, but are still a meaningful improvement of the state of the art on a challenging task.

**Weaknesses:**

This method has several limitations, although most of these limitations are shared by many other works in this area:
- the reliance on point cloud voxelization means that the entire scene must be completely observed up front and small enough to be voxelized. this makes it unusable for e.g. autonomous driving / robot navigation in large scenes.
- I am skeptical that the neural rendering objective can achieve stronger forms of generalization. "New position", "larger object" and "distractors" are all fairly weak forms of generalization, and I suspect the method would not be able to generalize to new environments or objects that have the same semantics but significantly different appearance from the training set. Ideally Stable Diffusion can help bridge this gap, but I assume the authors tried tougher OOD tasks and the method didn't work.
- the gap between the final checkpoint and the best checkpoint suggests to me that the training for these models is still quite unstable and requires a lot of tuning. I do see that this is a weakness of all the baselines too.
- going from a voxel grid to its radiance field is a little strange. the voxel grid already stores RGB and occupancy, so the 3D encoder basically has to learn the identity function and only recover view dependence to turn it into a radiance field. so I suspect the NeRF (RGB) objective barely contributes to learning at all. maybe the authors can verify / refute this.
- the method seems quite general but the types of environments/tasks explored in this paper are fairly constrained. it would have been nice to see some more diverse environments / agents

**Questions:**

- I don't particularly like the name NeRFuser because it sounds like a method for improving/fusing NeRFs. I also feel like the neural rendering component here is not the most important part of the pipeline, and the 2D teacher is at least equally as important. See if you can come up with a better name.
- I am not entirely sure what is the pre-training / training / test set for the real robot experiment. I think it would be interesting if you trained only on RLBench and tested on real robot tasks, but my understanding is that this combination was not tried. it would also be nice to repeat the real robot results several times to report stdev in the table
- can you describe exactly how the stable diffusion features are extracted? you feed in the input image, encode it, add noise (to which timestep?), then denoise (in 1 step)?

---

> ### Author Response · Authors · 2023-11-21
> **Author Response to Reviewer Pcik (1/2)**
>
> We thank the reviewer for the thoughtful comments. We address each of your comments in the following.
>
> ---
>
> **Q1:** The reliance on point cloud voxelization means that the entire scene must be completely observed up front and small enough to be voxelized. This makes it unusable for e.g. autonomous driving / robot navigation in large scenes.
>
> **A1:** Thanks for pointing this out. We agree this is a main challenge for voxel-based methods in large scenes. However, there are several ways to solve this problem. Firstly, our method is quite flexible and can be extended to using a point-based network to process point cloud directly and construct a 3D feature volume from which we sample point features for neural rendering and policy learning. This helps avoid point cloud voxelization and can be more efficient in large scenes, which is a promising research direction. In addition, previous work [1] has proposed to use occupancy networks to drastically reduce the memory requirements and increase the voxel grid resolution. This enables us to use a high-resolution voxel grid and effectively accommodate larger scenes.
>
> ---
>
> **Q2:** I am skeptical that the neural rendering objective can achieve stronger forms of generalization. "New position", "larger object" and "distractors" are all fairly weak forms of generalization, and I suspect the method would not be able to generalize to new environments or objects that have the same semantics but significantly different appearance from the training set. Ideally Stable Diffusion can help bridge this gap, but I assume the authors tried tougher OOD tasks and the method didn't work.
>
> **A2:** Thank you for your question. Generalizing to novel objects or those with significantly different appearances can be challenging when training is limited to our existing datasets. However, the NeRFuser's design, which separates 3D semantic training from policy learning, offers the potential to train from large-scale multi-view datasets and achieve high generalizability as foundation models. This is because previous works require the datasets to contain both actions and multi-view information and only a scarcity of data can meet the requirements. On the contrary, in NeRFuser, the datasets for neural rendering and policy learning are necessarily required to be the same, which allows us to use a multi-view dataset without actions (e.g. SUN3D [1]) and a demonstration dataset that is not multi-view (e.g. RT-X dataset [2]) to train the NeRFuser.
>
> ---
>
> **Q3:** The gap between the final checkpoint and the best checkpoint suggests to me that the training for these models is still quite unstable and requires a lot of tuning. I do see that this is a weakness of all the baselines too.
>
> **A3:** Thank you for pointing this out. This gap shows that multi-task training is still challenging for all manipulation methods. This is a question we will investigate in our future work.
>
> ---
>
> **Q4:** Going from a voxel grid to its radiance field is a little strange. The voxel grid already stores RGB and occupancy, so the 3D encoder basically has to learn the identity function and only recover view dependence to turn it into a radiance field. so I suspect the NeRF (RGB) objective barely contributes to learning at all. maybe the authors can verify/refute this.
>
> **A4:** We have done experiments training NeRF without RGB objective during the pre-training phase. We found the RGB objective significantly affected the stability of the NeRF Pre-training process. Only using feature embedding objective without RGB can easily lead to model collapse in the early training stage.
>
> ---
>
> **Q5:** The method seems quite general but the types of environments/tasks explored in this paper are fairly constrained. it would have been nice to see some more diverse environments / agents
>
> **A5:** Thank you for your suggestion. For simulation, we adopt RLBench, which contains a standard simulation environment and benchmarked tasks that are used by previous work for fair comparison and we follow the same setting to show that NeRFuser outperforms baseline significantly. For real robot experiments, we include 5 challenging real-robot tasks to evaluate the performance of the NeRFuser and the baselines. The number is greater than the 3 real-robot tasks used in the GNFactor.
>
>
> ---

---

> > ### Author Response · Authors · 2023-11-21
> > **Author Response to Reviewer Pcik (2/2)**
> >
> > **Q6:** I don't particularly like the name NeRFuser because it sounds like a method for improving/fusing NeRFs. I also feel like the neural rendering component here is not the most important part of the pipeline, and the 2D teacher is at least equally as important. See if you can come up with a better name.
> >
> > **A6:** Thank you for pointing this out. Yes, we agree that there are multiple components that contribute to the flexibility and efficiency of this method. We have updated the name to DNAct, where D refers to Diffusion, N refers to NeRFs, and Act refers to action and policy.
> >
> > ---
> >
> > **Q7:** I am not entirely sure what is the pre-training / training / test set for the real robot experiment. I think it would be interesting if you trained only on RLBench and tested on real robot tasks, but my understanding is that this combination was not tried. it would also be nice to repeat the real robot results several times to report stdev in the table
> >
> > **A7:** For real robot experiments, we use three RealSense cameras to collect a multi-view dataset for the pre-training phase and we simultaneously collect the demonstration dataset (for the following policy learning phase) with the front RealSense camera. To evaluate the model, we do not require an additional dataset and directory to obtain action predicted by the policy MLP.
> > It is indeed an interesting direction to utilize the simulation datasets to pre-train the model. In fact, one of the contributions of NeRFuser is that it separates the neural rendering and the policy learning components, which allows the model to have the flexibility to train and benefit from two distinct datasets such as SUN3D [2] multi-view dataset for indoor scene understanding and RT-X large-scale trajectory dataset [3] for policy learning. For the real robot experiments, it is still very time-consuming and expensive to repeat evaluations multiple times. In future work, we hope to explore more efficient methods that might allow for such repetitions.
> >
> > ---
> >
> > **Q8:** Can you describe exactly how the stable diffusion features are extracted? you feed in the input image, encode it, add noise (to which timestep?), then denoise (in 1 step)?
> >
> > **A8:** Yes, we use RGB images and corresponding language instructions as input for the Stable Diffusion model. Specifically, we encode the language instruction with a pre-trained text encoder and extract the text-to-image diffusion UNet’s internal features by feeding language embedding and RGB images into the UNet without adding noise. The rich and dense diffusion features are then extracted from the last layer of the UNet blocks. We only require a single forward pass to extract visual features instead of going through the entire multi-step generative diffusion process.
> >
> >
> >
> > [1] Mescheder, et al. “Occupancy networks: Learning 3d reconstruction in function space” Proceedings of the IEEE/CVF conference on computer vision and pattern recognition 2019.
> >
> > [2]Xiao, Jianxiong, Andrew Owens, and Antonio Torralba. "Sun3d: A database of big spaces reconstructed using sfm and object labels." Proceedings of the IEEE international conference on computer vision 2013.
> >
> > [3] Padalkar, Abhishek, et al. "Open x-embodiment: Robotic learning datasets and rt-x models." arXiv preprint arXiv:2310.08864 (2023).
> >
> > ---
> >
> > Please do not hesitate to let us know if you have any additional comments.

---

> > > ### Comment · Reviewer_Pcik · 2023-11-21
> > > **Thanks authors**
> > >
> > > Thanks for the helpful responses! It seems that the other negative reviews hinge on whether this work is sufficiently novel and/or meaningfully improves over GNFactor and PerAct. Based on an admittedly quick read of those other papers, I believe this paper passes these bars so I maintain my accept rating.

---

> > > > ### Author Response · Authors · 2023-11-23
> > > >
> > > > Dear Reviewer Pcik,
> > > >
> > > >
> > > > As the end of the rebuttal period approaches, we would like to sincerely express our gratitude for your insightful review. If you have any additional comments or suggestions regarding our work, please do not hesitate to let us know. Your feedback is invaluable to us.
> > > >
> > > > Thank you once again for your time and attention.
> > > >
> > > >
> > > > Best,
> > > >
> > > > Authors of Paper 1182

---

### Official Review · Reviewer_ui5H · 2023-10-30

**Soundness:** 3 good
**Presentation:** 2 fair
**Contribution:** 2 fair
**Rating:** 3
**Confidence:** 4

**Summary:**

The paper presents a method for language-conditioned multi-task policy learning. NeRFuser pretrains a visual representation using novel view synthesis as an objective with volume rendering. The method uses stable diffusion to distill semantic features into a 3D voxel encoder. Then the method uses Denoising Diffusion Probabilistic Models to reconstruct action sequences to train the decoder which fuses the encoded text from CLIP, the NeRF encoder, and PointNext encoded point clouds. Finally the method learns a policy on top of the learned representation. The paper performs experiments on RLBench and a real-world tabletop environment. NeRFuser shows significantly improved success rate over two recent methods, GNFactor [1] and PerAct [2].

1. GNFactor: Multi-Task Real Robot Learning with Generalizable Neural Feature Fields.
2. Perceiver-actor: A multi-task transformer for robotic manipulation.

**Strengths:**

- The performance gains over GNFactor and PerAct are statistically significant (over 16% higher success rate across 10 tasks).
- The benchmarking experiments are extensive with 5 real-world tasks, 10 in-distribution simulated tasks, and 5 unseen simulated tasks.
- The method figure is clear and helps to understand all of the components of NeRFuser.

**Weaknesses:**

- The stated technical contributions have been proposed by previous works.
    - In the abstract the work claims as a contribution: “NeRFuser leverages neural rendering to distill 2D semantic features from foundation models.” However, this exact setup was proposed by GNFactor [1].
    - The other contribution is stated to be:  "We introduce a novel approach utilizing diffusion training to learn a
vision and language feature". This approach seems very similar to previous works Janner et al. [2] and Huang et al. [3]


1.  GNFactor: Multi-Task Real Robot Learning with Generalizable Neural Feature Fields.
2. Planning with diffusion for flexible behavior synthesis
3. Diffusion-based generation, optimization, and planning in 3d scenes

- Given the close similarity to GNFactor, it would be useful to have experiments to analyze which component of NeRFuser is providing the improved performance. The ablation study which changes out stable diffusion for other pretrained models is interesting, but doesn't show the merits of NeRFuser over GNFactor.
- A potential weakness of the method, albeit minor is that it requires a specialized setup during training with multiple calibrated cameras.

- Details are unclear, particularly in the methods section (see questions).

**Questions:**

- The quantitative numbers for GNFactor from their paper differ significantly (sometimes as much as 20% success rate) than the ones reported in NeRFuser, could you explain why the performance differs so much in your setup?
- It’s unclear in Table 3 what the commands drawer, meat, and safe mean. Slide and place make sense as natural language commands for the encoder, but the other words don’t seem like commands.
- Frames are passed to the system as pointclouds, but there is no mention of how the point clouds are obtained.

---

> ### Author Response · Authors · 2023-11-21
> **Author Response to Reviewer ui5H (1/3)**
>
> We thank the reviewer for the thoughtful comments. We address each of your comments in the following.
>
> ---
>
> **Q1:** The stated technical contributions have been proposed by previous works.
> In the abstract the work claims as a contribution: “NeRFuser leverages neural rendering to distill 2D semantic features from foundation models.” However, this exact setup was proposed by GNFactor [1].
> The other contribution is stated to be: "We introduce a novel approach utilizing diffusion training to learn a vision and language feature". This approach seems very similar to previous works Janner et al. [2] and Huang et al. [3]
>
> **A1:** We appreciate your concerns about the perceived similarities between our work and previous studies. To clarify, our approach differs from GNFactor and other related methods in two primary aspects:
>
> * Training: Unlike GNFactor, which employs joint training of NeRF and behavior cloning policy, our method uses pre-training with neural rendering. This distinct approach allows for better generalization and improved quality in neural rendering. While GNFactor’s joint optimization can lead to a conflict between objectives, affecting both rendering quality and semantic information utilization, our pre-training strategy avoids this issue. Our results demonstrate superior performance in multi-task robot learning compared to the Learning-from-Scratch baseline.
>
> * Data Utilization: GNFactor requires task-specific, multi-view images paired with expert demonstration data, which can be challenging to collect and scale. In contrast, our approach effectively leverages widely available, task-agnostic multi-view image data. This not only eases the data collection process but also enhances the versatility of our 3D foundation model, making it applicable to a broader range of downstream robotic tasks. Our method's efficacy is further demonstrated through its application to five previously unseen tasks, yielding better multi-task performance compared to baseline PerAct and GNFactor.
>
> Furthermore, it's important to highlight our incorporation of diffusion-based techniques for multi-modal representation. This aspect of our work marks a significant advancement over previous methods. We're one of the first to successfully apply diffusion training in multi-task manipulation, whereas earlier works, such as those by Janner et al. and Huang et al., are predominantly evaluated under single-task scenarios. By formulating representation learning as an action sequence reconstruction problem and integrating diffusion models, we achieve more stable training and eliminate the need for the time-consuming inference processes typical of these models. This multi-modal approach not only enhances our method's capabilities but also sets our work apart in the field of robotic task learning.
> In summary, our approach stands out due to its unique pre-training approach, effective use of multi-view data, and innovative application of diffusion models for multi-modal representation, all contributing to its enhanced generalization and performance in diverse robotic tasks.
>
> ---
>
> **Q2:** Given the close similarity to GNFactor, it would be useful to have experiments to analyze which component of NeRFuser is providing the improved performance. The ablation study which changes out stable diffusion for other pretrained models is interesting, but doesn't show the merits of NeRFuser over GNFactor.
>
> **A2:** In our ablation study, we highlight the effectiveness of pre-training 3D representations through neural rendering. This is evident in Table 4, which shows a significant performance gap between NeRFuser and the Learning from Scratch (LfS) approach. This is the key difference between the proposed NeRFuser and the joint learning paradigm of GNFactor.  Our method's success rate substantially decreases without pre-trained 3D representations. We further provide the results of pre-training with data from five unseen tasks. It also achieves better results than GNFactor, showcasing the efficacy of our pre-training approach.

---

> ### Author Response · Authors · 2023-11-21
> **Author Response to Reviewer ui5H (2/3)**
>
> **Q3:** A potential weakness of the method, albeit minor is that it requires a specialized setup during training with multiple calibrated cameras.
>
> **A3:** The proposed NeRFuser greatly improves the line of research that utilizes semantic information for robotic manipulation. [4] requires around 50 calibrated scenes for NeRF training after every action inference. GNFactor [1] leverages generalizable NeRF and does not need per-scene optimization as [4]; however, it requires paired multi-view images and action data for both neural rendering and policy learning, which greatly limits the flexibility of learning a more scalable model.
> NeRFuser, on the other hand, disentangles the neural rendering and the policy learning components, which allows for learning from task-agnostic multi-view datasets to construct semantic representation in a 3D space, and **the training datasets for the policy learning are not required to have multiple views**. This improvement significantly reduces the training and inference time since we only need to query the pre-trained 3D encoder once for the downstream tasks. NeRFuser not only learns a more generalizable policy but also is more time-efficient.
>
> ---
>
> **Q4:** The quantitative numbers for GNFactor from their paper differ significantly (sometimes as much as 20% success rate) than the ones reported in NeRFuser, could you explain why the performance differs so much in your setup?
>
> **A4:** This performance difference comes from different training settings. We use 50 demonstrations and 200K iterations for GNFactor training while they use 20 demonstrations and 100k in the paper. With more demonstration data, GNFactor achieves better results in our setup.
>
> ---
>
> **Q5:** It’s unclear in Table 3 what the commands drawer, meat, and safe mean. Slide and place make sense as natural language commands for the encoder, but the other words don’t seem like commands.
>
> **A5:** Thanks for pointing this out. The words 'drawer (D)', 'meat (N)', and 'safe (D)' are abbreviations for specific task names. 'drawer (D)' refers to the task 'Put the Item in Drawer with Distractor', 'meat (N)' corresponds to the task 'Meat off Grill in New Position', and 'safe (D)' stands for the task 'Put Money in Safe with Distractor'. We have provided videos for all 5 tasks at https://nerfuser.github.io/. These videos illustrate different language commands for each task, providing a clear understanding of what each task entails.

---

> ### Author Response · Authors · 2023-11-21
> **Author Response to Reviewer ui5H (3/3)**
>
> **Q6:** Frames are passed to the system as point clouds, but there is no mention of how the point clouds are obtained.
>
> **A6:** Thank you for pointing this out. In the simulation environment, we follow previous works like GNFactor and PerAct to obtain RGB and depth images and convert them to point clouds. In the real world, we use the point clouds estimated by RealSense cameras. For details regarding extracting point clouds from a RealSense camera, we invite the reviewer to refer to the documentation: https://dev.intelrealsense.com/docs/docs-get-started.
> We have also clarified in the updated manuscript that our point cloud is obtained from the RealSense camera for better clarity.
>
> Reference
>
> [1] Ze, Yanjie, et al. "Gnfactor: Multi-task real robot learning with generalizable neural feature fields." Conference on Robot Learning. PMLR, 2023.
>
> [2] Janner, Michael, et al. "Planning with diffusion for flexible behavior synthesis." arXiv preprint arXiv:2205.09991 (2022).
>
> [3] Huang, Siyuan, et al. "Diffusion-based generation, optimization, and planning in 3d scenes." Proceedings of the IEEE/CVF Conference on Computer Vision and Pattern Recognition. 2023.
>
> [4] Shen, William, et al. "Distilled Feature Fields Enable Few-Shot Language-Guided Manipulation." Conference on Robot Learning. PMLR, 2023.
>
>
> ---
>
> Please do not hesitate to let us know if you have any additional comments.

---

> > ### Author Response · Authors · 2023-11-22
> >
> > Dear Reviewer,
> >
> > Thank you for reviewing our paper. We hope our responses have addressed your concerns. If so, we kindly ask if you could reconsider your score for our work.
> > Since tomorrow is the deadline for the rebuttal period, should you have any further questions, please don’t hesitate to let us know.
> > Thank you for your time and effort.
> >
> > Best,
> > Authors of Paper 1182

---

> ### Comment · Reviewer_ui5H · 2023-11-22
> **Response to Author Rebuttal**
>
> I thank the authors for their thorough response. I still have significant concerns about the stated contributions in the context of other works which I will detail below.
>
> The first and primary contribution of the paper as claimed in the abstract, intro, and rebuttal, is:
> “We leverage NeRF as a 3D pre-training approach to learn a 3D representation that can unify both semantics and geometry. From distilling pre-trained 2D foundation models into 3D space, we learn a generalizable 3D semantic representation with commonsense priors from internet-scale datasets."
>
> However, GNFactor[1] used all of these elements in almost an identical way to NeRFuser/DNAct. In the rebuttal, the main difference as I understand it is that GNFactor trains the policy and multi-view objective jointly, while NeRFuser/DNAct pretrains with the multi-view objective, then trains the policy. Otherwise the voxel encoder, multi-view objective, and distillation of Stable Diffusion are all the same. For example, the objective function (2) in GNFactor is exactly the same as equation (2) in DNAct, and section 2.1-2.2 in DNAact is quite similar to section 3.1-3.2 in GNFactor. Given my current understanding of both works, I think claiming the NeRF/multi-view component of the model as a contribution is misleading as the multi-view part of DNAct is the almost same as GNFactor. I think explaining where the performance difference is coming from whether it be the use of the diffusion model for the policy or the non-joint training is important given the extreme similarity between GNFactor and DNAct/NeRFuser.
>
> 1. GNFactor: Multi-Task Real Robot Learning with Generalizable Neural Feature Fields
>
> Other more minor concerns I have:
>
> **A2**: "In our ablation study, we highlight the effectiveness of pre-training 3D representations through neural rendering. This is evident in Table 4, which shows a significant performance gap between NeRFuser and the Learning from Scratch (LfS) approach. This is the key difference between the proposed NeRFuser and the joint learning paradigm of GNFactor."
>
> **Response to A2**: Could not GNFactor just be pretrained with the multi-view objective exactly like NeRFuser?
>
> **A4**: "This performance difference comes from different training settings. We use 50 demonstrations and 200K iterations for GNFactor training while they use 20 demonstrations and 100k in the paper. With more demonstration data, GNFactor achieves better results in our setup."
>
> **Response to A4**: I think there should be justification for why the training setup was changed. Changing the experimental setup makes it hard to compare performance across other papers.
>
> In summary, my main concern is that the paper is claiming the use of NeRF with a multi-view objective for training a visual encoder as the main contribution which I think is misleading as GNFactor has proposed an almost identical method previously. I will keep my score the same unless my current understanding of the paper in the context of other works is mistaken.

---

> ### Author Response · Authors · 2023-11-23
> **Author Response to Reviewer ui5H**
>
> Thanks for the reviewer's detailed feedback. We understand your concern and would like to address each point raised, focusing particularly on the distinctions between our proposed NeRFuser/DNAct and GNFactor. We want to emphasize the fundamental differences between our pre-training approach and the joint training paradigm of GNFactor as follows. Additionally, we provide more details on key technique differences between the two approaches.
> * **NeRfuser/DNAct demonstrates more effective utilization in the 2D Diffusion feature than GNFactor.** Admittedly, both GNFactor and NeRfuser/DNAct distill 2D foundation models into 3D space via Neural rendering. However, GNFactor’s joint optimization leads to a significant conflict between neural rendering and action prediction objectives. We compare the novel view synthesis results between ours and GNFactor in the figure ([link](https://drive.google.com/file/d/1eiL8DMXwRKIyrTCHp3mKGuQEF0n-bjMg/view?usp=sharing)). The poor rendering performance of GNFactor reveals its significant limitation in learning high-quality visual representations. Our method instead shows much better rendering results than GNFactor, showcasing its strong capability in parsing all object instances and extracting scene semantics efficiently. We believe this is a key factor contributing to the superior performance of our method.
> * **NeRfuser/DNAct shows stronger scalability and generalization ability than GNFactor.** Our 3D pre-training approach has the potential of using widely accessible task-agnostic multi-view images to pre-train a 3D foundation model for various robotic tasks. We show this by using 5 unseen out-of-domain tasks for pre-training and still outperforming GNFactor. On the contrary, GNFactor requires extensive paired action-image data due to the nature of joint training. It’s difficult to apply GNFactor to larger-scale tasks with such inflexibility. This further validates the importance and scalability of our method in the real world since it’s very expensive to collect task-specific, multi-view images paired with expert demonstration data.
> * **Key Technique Differences between GNFactor and NeRFuser/DNAct.** We adopt a learning-from-scratch pointnext encoder to obtain task domain-adapted point feature and fuse it with sampled point feature from pre-trained volume feature. This facilitates the utilization of pre-trained 3D semantic features and ensures adaptability to a variety of tasks. In addition, instead of employing dense voxel representation for policy learning like GNFactor, we choose a more sparse and efficient point-based approach. We also use a deeper 3D voxel encoder than GNFactor (4.72M vs. 0.3M parameters) to better capture 3D semantics from voxel input.
>
> Lastly, we would like to emphasize that the contributions of NeRFuser/DNAct are not only the above-mentioned pre-training paradigm for generalized 3D semantic learning but also the proposed diffusion training. The proposed novel diffusion training is the first approach that can learn the inherent multi-modality in the multi-task demonstrations and it does not require the costly denoising phase for each action inference, which makes the proposed NeRFuser/DNAct more efficient and stable to train for complex multi-task manipulation.
>
> **Q:** Could not GNFactor just be pre-trained with the multi-view objective exactly like NeRFuser?
>
> **A:** As the reviewer mentioned, we actually conducted an experiment on GNFactor pre-training at the early stage of this paper. We found out that pre-training GNFactor shows worse performance than joint training. We attribute this to the inefficient optimization of dense voxel representation. This also led us to choose a more efficient approach as mentioned above in “Key Technique Differences”.
>
> **Q:** I think there should be justification for why the training setup was changed. Changing the experimental setup makes it hard to compare performance across other papers.
>
> **A:** It's important to note that varying the number of demonstrations and training iterations is a common practice tailored to specific goals and applications. In our work, we focus on assessing the performance of manipulation policies with larger datasets. This approach aligns with our objective to propose a flexible algorithm adaptable to a diverse range of datasets. Consequently, we increased the number of demonstrations from 20 to 50 and the training iterations from 10k to 20k to ensure comprehensive training and convergence of all models.
> Despite these modifications, we have taken steps to ensure that our experiments remain comparable with other approaches. We maintained the same training setting and evaluation across all methods tested. Furthermore, we utilized the GNFactor [official implementation](https://github.com/YanjieZe/GNFactor) to guarantee the reproducibility and reliability of our findings.
>
> ---
> We hope this clarifies your concerns and would be grateful if you could raise our score accordingly.

---

> > ### Author Response · Authors · 2023-11-23
> > **Author Response to Reviewer ui5H**
> >
> > Dear reviewer ui5H,
> >
> > We would greatly appreciate it if you could review our recent response. We hope that we have addressed your previous concerns. We actively stand by for the last hours of the discussion phase.

---

### Official Review · Reviewer_3vtu · 2023-11-01

**Soundness:** 3 good
**Presentation:** 2 fair
**Contribution:** 3 good
**Rating:** 6
**Confidence:** 4

**Summary:**

This method integrates neural rendering pre-training (with diffusion-based foundation model) and diffusion training to learn a unified 3D representation for multi-task robotic manipulation in complex environments. The 3D encoder is pre-trained using NeRF to synthesise novel views, predicting corresponding semantic features from Stable Diffusion, language-2D foundation model. This pre-training representation equips the policy with out-of-distribution generalisation ability. The paper shows clear performance improvements compared with previous methods and baselines.

**Strengths:**

1. Overall architectural choices are reasonable and intuitive with clear reasoning.
 - Denoising objective the authors adopted is well-known to have a good performance in terms of representation learning
 - The authors utilize distillation of Stable Diffusion to improve generalization, which can be clear motivation.
- The above things are well combined in two-phase framework.
2. The authors provide abundant analysis (generalization, ablation on components of the framework) in simulation and real world.

**Weaknesses:**

- One major concern is a lack of explanation about how the foundation model (Stable Diffusion) is integrated in Sec 2.2. I couldn't find what F(r) exactly is. Are per-pixel features from Stable Diffusion? Then, how exactly was the feature derived? What is the language condition? What was the timestep set to? Which layer of the Stable Diffusion U-Net's features did the authors utilize? Are these settings sensitive to changes in hyper parameters? Utilizing features from the Text-to-Image diffusion model or distilling the information contained in the diffusion model into 3D representations like NeRF is in itself a significant area where active research is being conducted.  I think a detailed explanation, reasoning, and related experiments on this are needed. If I've misunderstood something, I'd appreciate it if the authors could explain further.

- It is just a minor thing, but Table 4 is ambiguous. I think it would be good to supplement it a little more so that it's easy to understand.

**Questions:**

1. Please see weaknesses.
2. I'm quite interested in the results shown in Figure 5. Could you explain the results? I couldn't find a paragraph mentioning Figure 5. Could you show more visualization results in other settings (unseen, out-of-distribution, without distillation of vision foundation models)?

I'm willing to raise my rating if my concerns are addressed well.

---

> ### Author Response · Authors · 2023-11-21
> **Author Response to Reviewer 3vtu (1/2)**
>
> We thank the reviewer for the thoughtful comments. We address each of your comments in the following.
>
> ---
>
> **Q1:** One major concern is a lack of explanation about how the foundation model (Stable Diffusion) is integrated in Sec 2.2. I couldn't find what F(r) exactly is. Are per-pixel features from Stable Diffusion? Then, how exactly was the feature derived? What is the language condition? What was the timestep set to? Which layer of the Stable Diffusion U-Net's features did the authors utilize? Are these settings sensitive to changes in hyper parameters?
>
> **A1:** Yes, F(r) is denoted as a ground truth feature map extracted from the Stable Diffusion model. We use RGB images and corresponding language instructions as input for the Stable Diffusion model. Specifically, we encode the language instruction with a pre-trained text encoder and extract the text-to-image diffusion UNet’s internal features by feeding language embedding and RGB images into the UNet without adding noise. The rich and dense diffusion features are then extracted from the last layer of the UNet blocks. We only require a single forward pass to extract visual features instead of going through the entire multi-step generative diffusion process. The feature extraction process has no hyperparameters involved.
>
> ---
>
> **Q2:** It is just a minor thing, but Table 4 is ambiguous. I think it would be good to supplement it a little more so that it's easy to understand.
>
> **A2:** Thank you for pointing this out. We have added a detailed introduction to the caption to indicate that
> LfS indicates a learning-from-scratch baseline without pre-training.
> Task-agnostic pre-training indicates that the pre-training phase is trained with out-of-distribution data from 5 unseen tasks.
>
> ---

---

> > ### Author Response · Authors · 2023-11-21
> > **Author Response to Reviewer 3vtu (2/2)**
> >
> > **Q3:** I'm quite interested in the results shown in Figure 5. Could you explain the results? I couldn't find a paragraph mentioning Figure 5. Could you show more visualization results in other settings (unseen, out-of-distribution, without distillation of vision foundation models)?
> >
> > **A3:** Thanks for your interest. Figure 5 shows the novel view synthesis results of NeRFuser on both RGB and diffusion features in the real world. By distilling the 2D stable diffusion model into 3D space, we learn a 3D semantic representation and visualize rendered 2D feature maps. It shows NeRFuser parses all object instances and extracts scene semantics well. In our revised manuscript, we have also added the visualization of 3D point features extracted from a pre-trained 3D encoder using out-of-distribution data in Figure 6.
> >
> > ---
> >
> > Please do not hesitate to let us know if you have any additional comments.

---

> > > ### Author Response · Authors · 2023-11-22
> > >
> > > Dear Reviewer,
> > >
> > > Thank you for reviewing our paper. We hope our responses have addressed your concerns. If so, we kindly ask if you could reconsider your score for our work.
> > > Since tomorrow is the deadline for the rebuttal period, should you have any further questions, please don’t hesitate to let us know.
> > > Thank you for your time and effort.
> > >
> > > Best,
> > > Authors of Paper 1182

---

> ### Comment · Reviewer_3vtu · 2023-11-23
> **Reply**
>
> Thank you for the reply. It has eased my worries to some extent. There's not much time left, but I have an additional question about Answer 1 (A1). From my personal experience, when robot instruction rather than image caption-like text was injected into the language condition part of Stable Diffusion, it didn't seem to have a significant impact. In the case of this paper, is the effect of injecting instructions into Stable Diffusion significant in terms of "language-action alignment"? Or does it tend to rely more on the language condition added after PointNext?

---

> > ### Author Response · Authors · 2023-11-23
> > **Response to reviewer 3vtu**
> >
> > Thanks for your valuable feedback. The language input for the stable diffusion model is significant in extracting language-aligned diffusion features. While this integration is beneficial, it is not the primary factor in our policy’s performance. Indeed, our policy is mainly conditioned on the language instruction added after PointNext. This language conditioning is essential for multi-task performance, as it equips the agent with a clearer understanding of various tasks and enhances its ability to respond robustly in diverse scenarios.
> >
> > Hope this is helpful and addresses your concern.

---

> ### Comment · Reviewer_3vtu · 2023-11-23
> **Reply**
>
> Thank you for the prompt response. Contrary to previous works utilizing SD(Stable Diffusion) in the robot field that have tackled data augmentation part with SD, this paper improves performance with a simple method that injects features extractable by a single feedforward pass of the text-to-2D diffusion model. Technical novelty aside, this discovery will be a valuable one in the robot field as it is a simple yet effective way to enhance performance. (though this has already been discovered in other fields such as segmentation)
>
> However, as Reviewer ui5H pointed out, in terms of overall architecture, this paper slightly lacks novelty. In this case, the main contribution of this paper would be the pretraining method using the pretrained diffusion model's feature. However, from this perspective, it still feels lacking in detailed explanation and analysis.
>
> Therefore, this paper still requires improvement. Nonetheless, considering that these points are already reflected in the ratings of other reviewers and the major contribution still exists, I lean towards acceptance, changing my rating to 'borderline accept'.

---

> > ### Author Response · Authors · 2023-11-23
> > **Response to reviewer 3vtu**
> >
> > Thank you for providing us with your valuable feedback. We appreciate your time and effort in reviewing our paper and responses. We will include more detailed explanation and analysis of our 3D pre-training using diffusion feature in our final manuscript.

---

### Official Review · Reviewer_bCxa · 2023-11-02

**Soundness:** 3 good
**Presentation:** 3 good
**Contribution:** 2 fair
**Rating:** 5
**Confidence:** 3

**Summary:**

The paper proposes NeRFuser that utilizes volume rendering pre-training and diffusion processes to learn the inherent multi-modality in the multi-task demonstrations. NeRFuser developes a 3D encoder capable of providing 3D semantic information in pre-training phases. In order to integrates multi-modal features, NeRFuser formulate the representation learning as an action sequence reconstruction with DDPM. It outperforms baseline approaches with an improvement of over 30% in 10 RLBench tasks and 5 real robot tasks.

**Strengths:**

1.The proposed idea is reasonable and its framework is well designed.
2.The authors provide adequate experiments and visualization results.

**Weaknesses:**

1.The author did not clearly articulate the motivation and contributions of the paper.
2.In my opinion, the writing of the paper is confusing. For example, in the first paragraph of introduction, why are the issues raised considered important and in need of resolution? As a researcher in another field, I cannot determine if this is a consensus within the field or the personal opinion of the author.
3.Based on the author's statements in the second paragraph of the introduction and the proposed methodology, it gives me the impression that the two methods have been cleverly combined together, lacking novelty to some extent.

**Questions:**

1.In Table 1, there seems to be a significant discrepancy between the performance of PerAct and the results reported in the original paper. Is this difference due to variations in the calculation of metrics or because the PerAct used for comparison in the experiments did not fully converge?
2.In the original paper, the author argues that the extensive inference time of diffusion models is a significant drawback. However, why is this issue resolved by adding a policy network? Additionally, I did not find any results related to speed or inference time in the experimental section.

---

> ### Author Response · Authors · 2023-11-21
> **Author Response to Reviewer bCxa (1/3)**
>
> We thank the reviewer for the thoughtful comments. We address each of your comments in the following.
>
> ---
>
> **Q1:** The author did not clearly articulate the motivation and contributions of the paper.
>
> **A1:** Thank you for your suggestion regarding the clarity of the paper. We have improved the clarity of the introduction sections to emphasize and pinpoint the motivation and contributions of the proposed NeRFuser in the revised manuscript. Here is the summary of the motivation and contributions:
>
> **Motivation:**
>
> * To perform complex manipulation tasks in the unstructured real world, it is essential to have a comprehensive semantic and geometric understanding of the scene. Previous NeRF-based manipulation methods ([1][2]) mainly focus on learning geometric features but have no semantic understanding of complex scenes.
> * In addition, being able to distinguish different modes in demonstration data is important for multi-task imitation learning. Diffusion policy is a great tool for this but still requires substantial parameter tuning to achieve accurate action prediction. It also needs a time-consuming multi-step denoising process, which results in higher inference latency than non-diffusion behavior cloning methods.
>
> **Contributions:**
>
> * We leverage NeRF as a 3D pre-training approach to learn a 3D representation that can unify both semantics and geometry. From distilling pre-trained 2D foundation models into 3D space, we learn a generalizable 3D semantic representation with commonsense priors from internet-scale datasets. We show that this pre-trained representation achieves strong out-of-distribution generalization ability. Unlike other NeRF-based methods ([1][2][3]), our method eliminates the need for task-specific in-domain datasets consisting of multi-view images paired with action data for 3D pre-training, which can be expensive to collect in the real world.
> * We introduce a novel approach utilizing diffusion training to learn a vision and language feature that distinguishes multi-modality from multi-task demonstrations. By utilizing a feature-conditioned noise predictor and reconstructing the action sequences from different tasks via the diffusion process, the model is capable of distinguishing different modalities and thus improving the robustness and the generalizability of the learned representation. The additionally learned policy network bypasses the need for the time-consuming denoising phase and makes multi-task learning more stable.
> * NeRFuser significantly surpasses strong baselines GNFactor and PerAct by over 30% in 10 RLBench tasks and 5 real robot tasks, illustrating the strong generalizability of NeRFuser.
>
> ---
>
> **Q2:** In my opinion, the writing of the paper is confusing. For example, in the first paragraph of introduction, why are the issues raised considered important and in need of resolution? As a researcher in another field, I cannot determine if this is a consensus within the field or the personal opinion of the author.
>
> **A2:**  First, both semantic and geometric understanding are quite necessary for complex real-world environments ([1][2][3]). For example, in a challenging scene (e.g., partial occlusion, various object shapes, and spatial relationship), comprehending the 3D structure of the scene is important for accurate manipulation and avoiding collision. Understanding the semantics and functionality of various objects is also crucial to a robust and generalizable policy in diverse scenes. Second, modeling the multi-modality of demonstration data has been one of the key challenges in behavior cloning literature ([4][5]). Without identifying the inherent multi-modality in the multi-task demonstrations, the learned policy is prone to overfitting to a small number of demonstration data and difficult to generalize to new scenes with novel objects and arrangements presented.
>
> ---

---

> ### Author Response · Authors · 2023-11-21
> **Author Response to Reviewer bCxa (2/3)**
>
> **Q3:** Based on the author's statements in the second paragraph of the introduction and the proposed methodology, it gives me the impression that the two methods have been cleverly combined together, lacking novelty to some extent.
> **A3:** We would like to clarify this point and emphasize our novelty and contributions as mentioned in A1 that NeRFuser
> * Explores a novel direction of leveraging neural rendering as a 3D pre-training approach to learn a 3D semantic representation. It achieves strong out-of-distribution generalization ability. This pre-training paradigm also disentangles multi-view datasets for neural rendering and actions datasets for policy learning, which greatly enhances the flexibility of learning from a variety of datasets as well as provides the potential of training a 3D foundation model for robotic manipulation. Previous work either requires datasets that are simultaneously multi-view and action-equipped or does not include 3D semantic information at all.
> * The diffusion-based representation learning is a novel method for learning multi-modal representation for observations. While previous diffusion-based policy learning has been successful in modeling action distribution for decision-making problems, they are still limited in multi-task settings with more complex environments. We instead focus on learning a multi-modal representation across multiple challenging tasks. This is achieved by introducing a novel diffusion objective to optimize the learned representation to discriminate various modes from the multi-task demonstration. This proposed method enables a much shorter action inference time than learning a diffusion policy and meanwhile benefits from the diffusion training and becomes more generalizable.
> * We want to emphasize again that our method achieves over 30% success rate improvement compared with strong baselines. This substantial improvement showcases the effectiveness of our method.
>
> ---
>
> **Q4:**  In Table 1, there seems to be a significant discrepancy between the performance of PerAct and the results reported in the original paper. Is this difference due to variations in the calculation of metrics or because the PerAct used for comparison in the experiments did not fully converge?
> **A4:**  In PerAct's experiments, they employ 10 and 100 demonstrations for 18 tasks. In our experiment, we apply the same training setting for both peract and our approach by using 50 demonstrations for 10 tasks. To ensure a fair and balanced comparison, we have applied equivalent training configurations for both PerAct and NeRFuser in the paper.
>
> ---

---

> > ### Author Response · Authors · 2023-11-21
> > **Author Response to Reviewer bCxa (3/3)**
> >
> > **Q5:** In the original paper, the author argues that the extensive inference time of diffusion models is a significant drawback. However, why is this issue resolved by adding a policy network? Additionally, I did not find any results related to speed or inference time in the experimental section.
> >
> > **A5:** To predict an action with a diffusion policy, it is required to go through a “denoising” process so that an accurate action signal can be constructed from a pure Gaussian noise. Previous work [4] takes around 100 denoising steps for this process and each step queries the noise predictor network once. The proposed method utilized the diffusion training to learn the multi-modal representation for observations. Notably we directly predict the corresponding action with a policy MLP network instead of the diffusion model. The size of the policy network is less than 10% of that of the noise predictor network and we only need to query it once for every action inference, which makes the proposed method over 10 times faster in action inference and still benefit from the multi-modality obtained from the diffusion training. Since it involves different implementations such as point sampling empirically, we still observe more than 15 times speedup compared to the diffusion policy.
> >
> > Reference
> > [1] Driess, Danny, et al. "Reinforcement learning with neural radiance fields." Advances in Neural Information Processing Systems 35 (2022): 16931-16945.
> >
> > [2] Li, Yunzhu, et al. "3d neural scene representations for visuomotor control." Conference on Robot Learning. PMLR, 2022.
> >
> > [3] Ze, Yanjie, et al. "Gnfactor: Multi-task real robot learning with generalizable neural feature fields." Conference on Robot Learning. PMLR, 2023
> >
> > [4] Chi, Cheng, et al. "Diffusion policy: Visuomotor policy learning via action diffusion." arXiv preprint arXiv:2303.04137 (2023).
> >
> > [5] Florence, Pete, et al. "Implicit behavioral cloning." Conference on Robot Learning. PMLR, 2022.
> >
> > ---
> >
> > Please do not hesitate to let us know if you have any additional comments.

---

> > > ### Author Response · Authors · 2023-11-22
> > >
> > > Dear Reviewer,
> > >
> > > Thank you for reviewing our paper. We hope our responses have addressed your concerns. If so, we kindly ask if you could reconsider your score for our work.
> > > Since tomorrow is the deadline for the rebuttal period, should you have any further questions, please don’t hesitate to let us know.
> > > Thank you for your time and effort.
> > >
> > > Best,
> > > Authors of Paper 1182

---

> > > > ### Author Response · Authors · 2023-11-23
> > > > **Author Response to Reviewer bCxa**
> > > >
> > > > Dear reviewer bCxa,
> > > >
> > > > Since the end of the rebuttal period approaches, we kindly ask if you could reconsider your score for our work. We hope that our responses have addressed your previous concerns. We actively stand by for the last hours of the discussion phase.

---

### Author Response · Authors · 2023-11-21

We thank all reviewers for their thoughtful comments and insights. We have revised our manuscript based on your feedback, and we have responded to each of your comments.

Summary of revisions:
* Improve the clarity of the introduction section for clearer motivation and contribution.
* Add implementation details regarding stable diffusion feature extraction for better reproducibility.
* Add a more detailed explanation regarding Table 4 and Figure 5.
* Add visualization of pre-trained 3D point features in Figure 6.
* Update the method name from NeRFuser to DNAct, where D refers to Diffusion, N refers to NeRFs, and Act refers to action and policy.
* Update the caption of Table 4 to give a more detailed explanation such as the abbreviations (LfS).


We would like to highlight the key contributions of our proposed NeRFuser:

* **Novel 3D Pre-Training for Policy Learning:** Our approach leverages NeRF for 3D pre-training to learn a unified 3D representation of both semantics and geometry. This pre-trained representation exhibits strong generalization abilities even in unseen scenarios. Unlike previous NeRF-based methods that required extensive multi-view image collection for each action, our approach has stronger scalability by eliminating the need for extensive, action-specific image datasets. NeRFuser shows the possibility of effectively utilizing widely available, task-agnostic, multi-view image data for pretraining a versatile 3D foundation model applicable to a diverse range of downstream robotic tasks.
* **Novel Diffusion Training:** We introduce a diffusion training approach that effectively distinguishes the multi-modality inherent in multi-task demonstrations. Additionally, NeRFuser features a significantly reduced action inference time compared to a pure diffusion policy. This efficiency is due to its more compact network architecture, which is 10% smaller in parameters, and requires far fewer inferences (only 7%). Empirically, the proposed approach is more than 15 times faster than the diffusion policy.
* **Performance:** It surpasses the baseline GNFactor and PerAct by over 30% in both simulation and real-robot experiments, marking a substantial advancement in performance.


Again, we thank the reviewers for their constructive feedback. We hope that all comments have been addressed, but are happy to address any further comments from reviewers.


Best,

Authors of Paper 1182

---

### Meta-Review · Area_Chair_2YNT · 2023-12-03

**Metareview:**

The paper received divergent ratings (8,6,5,3). The reviewers raised several concerns such as clarity of the paper, contributions already proposed by prior work, proper ablation study to show the merits of NeRFuser over GNFactor, requiring calibrated cameras, etc. Some of the reviewers had discussions with the authors. Also, there was a discussion between the AC and the reviewers (not visible to authors). The main outstanding issue is the similarity to GNFactor. The main contribution of this paper (sections 2.1 and 2.2) is from the GNFactor paper. The difference with GNFactor is that this work pretrains and freezes the encoder, whereas GNFactor trains it jointly. The AC checked the paper, the reviews, the responses and the discussion. The paper needs revision to make the contributions clear and properly differentiate them with prior work. Hence, rejection is recommended.

**Justification For Why Not Higher Score:**

The main issue is the similarity with prior work and lack of novelty as expressed by reviewers 3vtu and ui5H.

**Justification For Why Not Lower Score:**

N/A

---

### Decision · Program_Chairs · 2024-01-16

Reject